# Similar Risk of Re-Revision in Patients after One- or Two-Stage Surgical Revision of Infected Total Hip Arthroplasty: An Analysis of Revisions in the Swedish Hip Arthroplasty Register 1979–2015

**DOI:** 10.3390/jcm8040485

**Published:** 2019-04-10

**Authors:** Karin Svensson, Ola Rolfson, Johan Kärrholm, Maziar Mohaddes

**Affiliations:** 1Department of Orthopaedics, Sahlgrenska University Hospital, 431 80 Mölndal, Sweden; ola.rolfson@vgregion.se (O.R.); johan.karrholm@vgregion.se (J.K.), maziar.mohaddes@gmail.com (M.M.); 2Department of Orthopaedics, Institute of Clinical Sciences, Sahlgrenska Academy, University of Gothenburg, 431 80 Mölndal, Sweden; 3The Swedish Hip Arthroplasty Register, 413 45 Gothenburg, Sweden

**Keywords:** hip arthroplasty, revision surgery, infection

## Abstract

Late chronic infection is a devastating complication after total hip arthroplasty (THA) and is often treated with surgery. The one-stage surgical procedure is believed to be the more advantageous from a patient and cost perspective, but there is no consensus on whether the one- or two-stage procedure is the better option. We analysed the risk for re-revision in infected primary THAs repaired with either the one- or two-stage method. Data was obtained from the Swedish Hip Arthroplasty Register and the study groups were patients who had undergone a one-stage (n = 404) or two-stage (n = 1250) revision due to infection. Risk of re-revision was analysed using Kaplan–Meier analysis with log-rank test and Cox regression analysis. The cumulative survival rate was similar in the two groups at 15 years after surgery (*p* = 0.1). Adjusting for covariates, the risk for re-revision due to all causes did not differ between patients who were operated on with the one- or two-stage procedure (Hazard Ratio (HR) = 0.9, 95% Confidence Interval (C.I.) = 0.7–1.2, *p* = 0.5). The risk for re-revision due to infection (HR = 0.7, 95% C.I. = 0.4–1.1, *p* = 0.2) and aseptic loosening (HR = 1.2, 95% C.I. = 0.8–1.8, *p* = 0.5) was similar. This study could not determine whether the one-stage method was inferior in cases when the performing surgeons chose to use the one-stage method.

## 1. Introduction

The demand for primary total hip arthroplasties (THA) has steadily increased during the last four decades. According to the Swedish Hip Arthroplasty Register (SHAR,) approximately 17,000 primary THAs are performed annually in Sweden. Prosthetic joint infection (PJI) is a feared complication after joint replacement surgery. With an incidence rate of 1–2%, PJI is one of the most common reasons for revision [1,2,3]. Orthopaedic infections imply great suffering for the affected patient, with high emotional distress and functional disability, and, in a few patients, PJI may lead to a fatal outcome [4,5,6]. Postoperative infections also create an economic burden in terms of high healthcare and societal costs [7]. The cost of managing an infected THA is estimated to be 4.8 times higher than the cost of performing a primary THA [8]. With a suggested increase in infection incidence, an ageing population, and increasing demands for THA, PJI is expected to become an even greater problem in the future [9].

There is currently an ongoing debate on some aspects of treatment for PJI [10]. Patients who present with an acute onset of infection typically undergo debridement and receive targeted antibiotics, in hope of retaining the prosthesis—a process also known as the Debridement, Antibiotics, and Implant Retention (DAIR) procedure. When DAIR is not successful and in patients presenting with late chronic infections, the surgical revision procedure aims to remove the infected prosthesis. This can either be done as a one- or two-stage revision and, more rarely, as a definitive extraction of the prosthesis.

In Sweden, the majority of infected THAs, after failed DAIR or presenting with a late onset infection, are revised using the two-stage procedure. According to the SHAR, 82% of the revisions between 1979 and 2015 were two-stage revisions. The two-stage revision has traditionally been regarded as the best method for infection control. However, to date, there have been no randomised clinical trials validating this. Single-centre case series and systematic reviews on the matter have found that the rate of reinfection is similar in comparisons between one- and two-stage revision surgery [11,12,13,14,15]. In addition, the one-stage method is regarded as the more advantageous in terms of cost, functional outcome, and patient experience [8,16,17].

In this study we performed an analysis of all first-time revisions performed due to PJI, reported to the SHAR during years 1979–2015. We evaluated if there was a higher risk of re-revision between the one- and the two-stage methods due to all causes, recurrent infection, or aseptic loosening after the revision of an infected primary THA.

## 2. Experimental Section

Patients revised with either a complete (all components removed) one- or two-stage procedure, due to infected primary THA reported to the SHAR during the years 1979–2015, were included in the study. The SHAR was started in 1979 and all orthopaedic units in Sweden performing revision hip arthroplasties report to this register. The majority (98%) of primary THAs in Sweden are reported to the register. By the end of 2015, it held information on 44,579 revision procedures, of which 34,299 were first-time revisions.

Baseline information on revisions are reported to the SHAR by local coordinators at each hospital. Copies of the medical records are sent to the SHAR from the participating orthopaedic unit for validation of inputted data and completion of other relevant variables. Mortality data on patients was obtained using the Swedish Tax Agency, which is linked to the SHAR. In 2000, Söderman et al. studied the validation of the SHAR and found that an average of 94% of all the revisions nationwide were reported [18]. Another validation was done in 2014 and it found that 78% of all revisions due to infection were reported to the SHAR [19]. In the SHAR, revision procedures are defined as the either the exchange of prothesis parts or the entire prosthesis, or, alternatively, the full extraction of the prosthesis. In this study, revision was defined as the exchange of the entire prosthesis, and each THA was studied separately in cases where patients had bilateral PJI.

The SHAR includes information on age, sex, surgery side, diagnosis, and details regarding the procedure, such as implant type, fixation method, and surgical approach. Patients who were revised with a one- or two-stage procedure for PJI after a primary THA due to osteoarthritis were identified in the SHAR.

During the years 1979–2015, 1654 first-time revisions due to infection after primary THA were reported to the SHAR. These revisions were performed either as two-stage (n = 1250) or one-stage (n = 404) revision procedures. The American Society of Anesthesiologists physical status classification (ASA score) and body mass index (BMI) were first registered in the SHAR in 2008. This data was available for 466 of the revision cases.

Re-revision, regardless of cause, was set as the primary outcome. Analysis of re-revision due to aseptic loosening and infection was also performed and used as secondary outcomes.

The difference in risk for re-revision between one- and two-stage revision surgery was analysed using the Kaplan–Meier survival analysis and log-rank test. The survival analysis was truncated at 15 years after surgery, after which the number of at-risk patients dropped below 100 cases. A Cox regression analysis, adjusted for sex, age, diagnosis (primary or secondary osteoarthritis), and method of fixation (cemented or non-cemented), was performed. A subgroup analysis of 466 cases was performed using Cox regression with the aforementioned variables as well as BMI, ASA score, and year of surgery.

The two-stage revision group was used as a reference. Hazard ratio (HR), 95% confidence interval (C.I.), and *p*-values were used to present the Cox regression analysis. Demographic data is presented as the mean and standard deviation (SD). Statistical significance was defined as a p-value of less than 0.05.

There was a significant difference in age (*p* = 0.01), where patients who had one-stage revision were a mean age of 70 (SD ± 10) years and patients who had a two-stage revision were a mean age of 68 (SD ± 10) years (Table 1). The time between the primary operation and the first revision was a mean of 3.1 (SD ± 3.7) years in the one-stage group and 4.1 (SD ± 4.4) years in the two-stage group (*p* < 0.001). The follow-up time was a mean of 10.9 (SD ± 9.4) years in the one-stage group and 7.9 (SD ± 6.7) years in the two-stage group (*p* < 0.001).

IBM SPSS Statistics version 22 was used for analysis. The study was approved by the Regional Ethical Review Board in Gothenburg, Sweden (entry number 271-14).

## 3. Results

The two-stage procedure has been used in at least 80% of revision cases, making it the most common revision method in Sweden during the past 25 years (Table 2). The number of one-stage procedures has, however, proportionally increased during 2010–2015 compared to 2000–2009 (Figure 1).

### 3.1 Re-Revision Due to All Causes

In the one-stage and two-stage groups, 83 cases (21%) and 259 cases (21%), respectively, were re-revised after their first revision due to all causes. Unadjusted, the cumulative survival rate was similar in the two groups at 15 years after surgery (*p* = 0.1), with a survival rate of 75% ± 5.4% for the one-stage group and 72% ± 3.6% for the two-stage group (Figure 2). Adjusted for the co-variates in a Cox regression model, there was no difference in the risk for re-revision between the one- and two-stage revision surgeries (HR = 0.9, 95% C.I. = 0.7–1.2, *p* = 0.5).

### 3.2 Re-Revision Due to Infection

Twenty-eight cases in the one-stage group and 107 cases in the two-stage group were re-revised due to infection. Unadjusted, the cumulative survival rate was similar in the one- and two-stage group at 15 years after surgery (*p* = 0.13), with a survival rate of 92% ± 3.1% and 89% ± 2.2%, respectively (Figure 3). Cox regression analysis showed that there was no difference in risk for re-revision due to infection (HR = 0.7, 95% C.I. = 0.4–1.1, *p* = 0.2).

### 3.3 Re-Revision Due to Aseptic Loosening

Thirty-five cases in the one-stage group and 84 cases in the two-stage group were re-revised due to aseptic loosening. The survival rate at 15 years after surgery was similar when using an unadjusted log-rank analysis (*p* = 0.9), with a cumulative survival rate of 88% ± 4.5% and 87% ± 3.1% in the one- and two-stage group, respectively (Figure 4). No difference in risk could be found between the groups when adjusted for covariates (HR = 1.2, 95% C.I. = 0.8–1.8, *p* = 0.5).

### 3.4 Subgroup Analysis

The subgroup analysis was performed to investigate whether there was a difference in outcome when adjusting for ASA score, BMI, and operation year. No difference in the risk for re-revision due to all causes could be found in the subgroup when these adjustments were made (HR = 0.7, 95% C.I. = 0.3–1.6, *p* = 0.4). The distribution of ASA score and BMI were similar (Table 3).

## 4. Discussion

The utilization of the one- or two-stage method in treating PJI remains controversial. The aim of this study was to compare the risk for re-revision due to all causes between the two methods based on data reported to the SHAR. Analyses of the risk for re-revision due to aseptic loosening and recurrent infection were also performed. Re-revision rates due to all causes were similar in the two groups.

This is the largest national observational study on the risk for re-revision between the one- and two-stage procedure. To our knowledge, there is only one other national study comparing the risk for re-revision due to all causes between the one- and two-stage procedure [20]. This study found an increased risk for re-revision due to all causes in the one-stage group when compared to the two-stage group (HR = 1.4), however, this difference was not significant (*p* = 0.2). Engesaeter et al. also found a two-fold increase in risk for re-revision due to infection after the one-stage procedure when compared to the two-stage procedure (*p* = 0.04) [20]. This is contradictory to our results. The two studies were performed in similar ways, but follow-up was set to two years in the study by Engesaeter et al. whereas ours was longer. The lack of consensus on diagnostic criteria for PJI could partly explain the difference between our results.

Based on the hazard ratio calculations and the Kaplan–Meier survival analyses, re-revision rates due to infection showed no significant difference between the two revision groups. Systematic reviews have not been able to find a significant difference in infection resolution between the one- and two-stage procedures [12,13,14]. A recent study, conducted by pooling individual participant data, could not find any difference in infection resolution between the methods either [21]. These findings support our results.

Risk for re-revision due to aseptic loosening was analysed as it can be hard to separate this from a low-grade infection. The relevance of positive cultures in the diagnosis of aseptic loosening is still not known [22]. We did not find any publications where re-revision risks due to aseptic loosening were compared between one- and two-stage revisions after PJI.

We could not analyse the influence of comorbidities in the total material since the ASA score was first registered in 2008 in the SHAR. The ASA score is used in clinical routine to assess patients’ physical status pre-operatively to predict morbidity and mortality. A high ASA score is associated with a greater risk of complication due to surgery [23]. To date, it is a general opinion that patients with severe or many comorbidities are more suitable for the two-stage than the one-stage approach. This may create a selection bias in a clinical setting, where patients with severe comorbidities are selected for the two-stage approach to a greater extent than for the one-stage approach. This might mean that the preconditions for infection resolution are better in the one-stage group, favouring this method. However, it could also be argued that some surgeons may prefer to use the one-stage method for patients with severe comorbidities to avoid exposing the patient to further surgery.

Some baseline demographic data differed significantly between the two groups in our study (Table 1). Patients in the one-stage group were older. Age as a risk factor for infection is inconsistent, but our findings may support the theory that surgeons select older patients for the one-stage procedure to avoid exposing them to further surgery. We cannot explain the difference in mean time from the primary procedure to the revision procedure between the two groups. The one-stage group had a longer follow-up time. The more part on one-stage revisions were performed early in the study period, and the mean time from the primary procedure to the revision procedure was shorter for this group, allowing for a longer follow-up time. Cementation was more common as type of fixation in the one-stage group. Older patients are more likely to receive cemented implants, which may explain this difference. However, the influence of cemented fixation on risk for infection has not been firmly established.

This study stretched over a long time period in which several changes in surgical technique, operative hygiene, microbial, and antimicrobial development have occurred. We therefore performed a sub-analysis including comorbidities, BMI, and year of revision to address these concerns and could not show any significant difference in risk of re-revision between the one- and two-stage groups.

The SHAR does not register the type of microbe causing PJI or details on antimicrobial treatment, which is another possible confounder. Bacteria with biofilm formation abilities have been found to be particularly challenging in the management of orthopaedic infections [24]. Therefore, the type of bacteria might influence the results if they significantly differ between the two groups. There are no clinical studies comparing the risk of recurrent infection after revision due to infection based on the biofilm formation abilities of different bacteria. It is, however, known that certain bacteria are more difficult to eradicate in cases of PJI. Since the infectious organism was unknown, we cannot determine if any selection of low virulent bacteria for the one-stage group influenced our results.

The diagnostics of PJI can be difficult. Several work groups have tried to formulate diagnostic criteria for PJI [25]. The criteria for PJI may vary between centres included in this study. However, there is a validation process in the SHAR where the case records are studied by two specially-trained coordinators at the SHAR after the baseline data has been inputted by local coordinators. We, therefore, think that the cases recorded as infections in the SHAR were revised due to infection and no other causes.

Some revisions due to infection performed in Sweden during 1979–2015 have not been registered in the SHAR. Lindgren et al. performed an external validation on the reporting of reoperations due to infection in primary THA to the SHAR [19]. They found that, during the period studied (2005–2008), 78% of revisions due to infection were reported. We assume that the distribution of non-reported cases does not differ between the one- and two-stage revisions.

Finally, our end-point was re-revisions. This means that cases still infected and not re-revised were not known to us.

Findings in this study should be interpreted in light of residual confounding factors not known to us, and with consideration of the possible influence of treatment and microbial development during the study’s time span on our results. In a clinical setting, each patient presents with different challenges, in terms of comorbidities and infections with different microbiological profiles. These data, not recorded in national registries, might have an impact on outcome after surgery.

Although there were limitations, we believe that our study, as the largest national observational study, with reporting on real life outcome after one- and two-stage revisions following infected THA is of interest for the orthopaedic community and adds to the existing literature.

## 5. Conclusions

There is lack of consensus on which revision method is best in the treatment of PJI. The one-stage method has recently gained more attention due to its possible advantages. Analyses from this nation-wide register study support the utilization of the one-stage procedure due to infection when the surgeon finds it suitable. There is need for further studies to validate the findings from this register study and to identify the patient, infection, or surgical factors that predict successful outcome following one- or two-stage revision in infected THA.

## Figures and Tables

**Figure 1 jcm-08-00485-f001:**
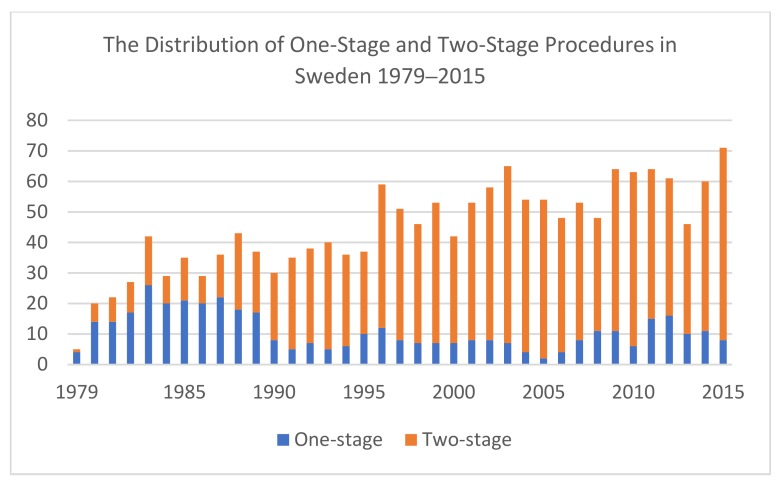
The distribution of one- and two-stage procedures in Sweden during 1979–2015.

**Figure 2 jcm-08-00485-f002:**
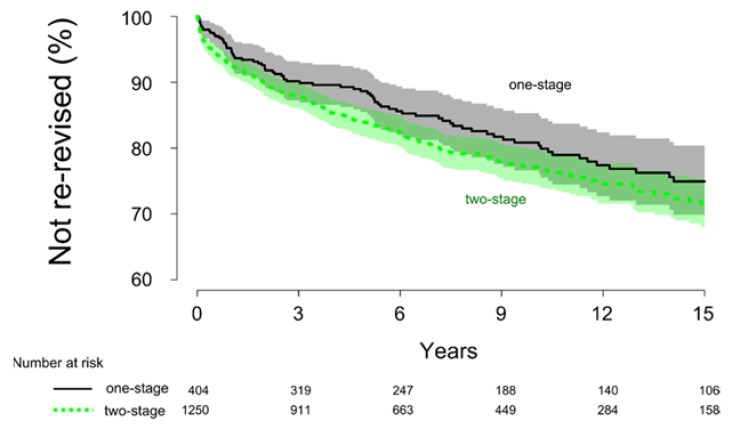
Kaplan-Meier survival analysis comparing the one- and two-stage procedure using re-revision due to all causes as an endpoint. The cumulative survival rate at 15 years after surgery was found to be similar between the two procedures (*p* = 0.13).

**Figure 3 jcm-08-00485-f003:**
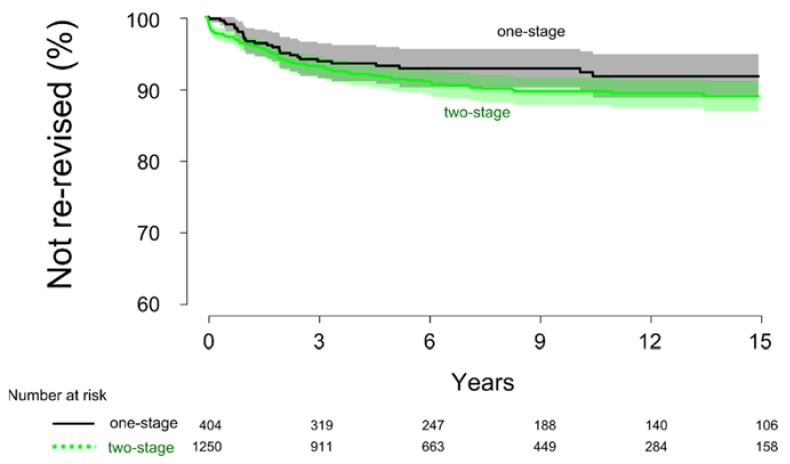
The cumulative survival rate at 15 years after surgery using re-revision due to infection as the endpoint. The survival rate was similar between the one- and two-stage procedures (*p* = 0.13).

**Figure 4 jcm-08-00485-f004:**
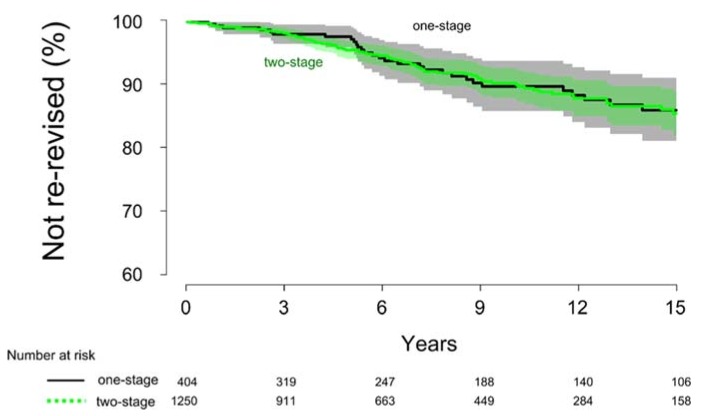
The cumulative survival rate at 15 years after surgery for the one- and two-stage group was similar (*p* = 0.87) when analysed for re-revision due to aseptic loosening.

**Table 1 jcm-08-00485-t001:** The demographics of the one-and two-stage revision groups.

Baseline Demographics in the One- and Two-Stage Revision Groups			
	One-Stage	Two-Stage	*p*-Value
**Sex** *n (%)*			0.838
Male	246 (60.9)	754 (60.3)	
Female	158 (39.1)	496 (39.7)	
**Age at reoperation** (yrs ± SD)	70 ± 10	68 ± 10	0.012
**Side** *n (%)*			0.825
Right	222 (55.0)	679 (54.3)	
Left	182 (45.0)	571 (45.7)	
**Diagnosis** *n (%)*			0.556
Primary osteoarthritis	314 (77.7)	964 (77.1)	
Secondary osteoarthritis	90 (22.3)	286 (22.9)	
*Inflammatory joint disease*	30	73	
*Fracture*	41	114	
*Sequelae after congenital disease*	9	41	
*Idiopathic necrosis of the femur caput*	8	33	
*Secondary post-traumatic arthritis*	2	19	
*Other*	0	6	
**Time to primary revision** (yrs ± SD)	3.1 ± 3.7	4.1 ± 4.4	<0.001
**Follow-up time** (yrs ± SD)	10.9 ± 9.4	7.9 ± 6.7	<0.001
**Type of fixation at the revision** *n (%)*			<0.001
Cemented	332 (82.4)	732 (58.6)	
Non-cemented	34 (8.4)	239 (19.1)	
Other (hybrid techniques)	37 (9.2)	279 (22.3)	

yrs = years, SD = standard deviation

**Table 2 jcm-08-00485-t002:** The one- and two-stage revision utilization from 1979–2015.

One- and Two-Stage Usage over the Years 1979–2015			
Years	One-Stage *n (%)*	Two-Stage *n (%)*	Total *n (%)*
1979–1989	193 (59.4)	132 (40.6)	325 (100.0)
1990–1999	75 (17.7)	350 (82.3)	425 (100.0)
2000–2009	70 (13.0)	469 (87.0)	539 (100.0)
2010–2015	66 (18.1)	299 (81.9)	365 (100.0)

**Table 3 jcm-08-00485-t003:** The distribution of the American Society of Anesthesiologists (ASA) score and body mass index (BMI) in registered cases, 2008–2015.

The Distribution of ASA Score and BMI in Registered Cases			
	One-Stage n (%)	Two-Stage n (%)	*p*-Value
**ASA-classification**			0.440
1 Healthy	11 (13.1)	36 (9.4)	
2 Mild systemic disease	44 (52.4)	216 (56.5)	
3 Severe disease	28 (33.3)	126 (33.0)	
4 Incapacitating disease	1 (1.2)	4 (1.0)	
**BMI-classification**			0.755
1 Underweight	0 (0.0)	38 (0.8)	
2 Normal weight	20 (25.6)	113 (32.0)	
3 Overweight	36 (46.2)	132 (37.4)	
4 Obese class I	12 (15.4)	71 (20.1)	
5 Obese class II	8 (10.3)	26 (7.4)	
6 Obese class III	2 (2.5)	8 (2.3)	

ASA score = American Society of Anesthesiologists score, BMI = body mass index

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
