# Peer review of "Similar Risk of Re-Revision in Patients after One- or Two-Stage Surgical Revision of Infected Total Hip Arthroplasty: An Analysis of Revisions in the Swedish Hip Arthroplasty Register 1979–2015"

_jcm, 2019, doi:10.3390/jcm8040485_

Reviewer 1 Report

The authors have done a thorough study of the risks of re-revision surgery with one- and two-stage revision of THA based on the Swedish Hip Arthroplasty Register.  The data is well presented and the argument that one-stage revision is on the same risk level as two-stage revision is well documented and sustained.  However, the reviewer has some reservations that are given below.

The authors fail to substantiate their study with similar studies in other countries which would support their thesis.

The authors' statement on p. 3, line 109-110, which states that number of one-stage procedures have been increasing during the last years is not supported by Figure 1 nor Table 2.

While the numbers for re-revision expressed in percentage is similar for one- and two-stage groups, the numbers themselves are significantly different for the two cases with two-stage revision far exceeding one-stage revision procedures.

The same is true for the cases of re-revision due to infection and aseptic loosening.

The fact that there are no significant differences in the percentage values between one-stage and two-stage does not prove convincingly that one-stage revision is better or sufficient. 

Minor English revision needed.

Author Response

Point 1: The authors fail to substantiate their study with similar studies in other countries which would support their thesis.

Response 1: To our knowledge, our study is the first register-based study of its size investigating the risk for re-revision after the one- or two-stage procedure for PJI. Engesaeter et al. performed a similar study in Norway, which is mentioned in the discussion. Otherwise, we have not been able to identify comparable studies in other countries. Most often, studies investigate either the one- or the two-stage procedure, and they are of a smaller study sample and do not have a nation-wide coverage. There are recent meta-analyses supporting a similar outcome between the one- and two-stage procedure. Also, there is an IPD-study which has pooled data from 44 cohort studies that does not show a statistical significant difference between the procedures. These studies support our thesis, and are mentioned in the manuscript. We would be more than happy to consider including other references suggested by the reviewer.

Point 2: The authors' statement on p. 3, line 109-110, which states that number of one-stage procedures have been increasing during the last years is not supported by Figure 1 nor Table 2.

Response 2: This statement was based on the percental increase between the years 2010-2015 compared to 2000-2009. We realize that this was not explained properly, and that this was difficult to interpret from the table and figure based on our current statement. Therefore, we decided to revise the statement to clarify that the one-stage procedure has proportionally increased during 2010-2015 compared to 2000-2009 (p. 3, line 110).

Point 3: While the numbers for re-revision expressed in percentage is similar for one- and two-stage groups, the numbers themselves are significantly different for the two cases with two-stage revision far exceeding one-stage revision procedures. The same is true for the cases of re-revision due to infection and aseptic loosening.

Response 3: This is true. However, we believe that it is of interest to report on how many of the one- and two-stage cases were re-revised.

Point 4: The fact that there are no significant differences in the percentage values between one-stage and two-stage does not prove convincingly that one-stage revision is better or sufficient. 

Response 4: We certainly agree that no firm conclusions can be drawn concerning choice of one- or two-stage procedure from our study, which is observational and most certainly suffers from selection bias. Our conclusions are based on the hazard ratio calculations and Kaplan Meier survival analyses we have performed. We have clarified this in the discussion section of the revised manuscript (p. 6 line 171). The conclusions are not based on the descriptive percentage values, as they provide limited information on the risk for re-revision which is the end-point of our study.

Reviewer 2 Report

In line 82 what do they mean when they say: "Therefore, this data was not made available for all revision cases identified for the current analysis". The fear in this study is that a significant number of cases were missed out of the analysis (as many as 25% in certain periods of the study. This could make a big difference to the results

The Survival Analysis method used is appropriate and presented very well in the results

In their Stratification, its clear that the two groups are not the same - two-stage revisions were younger (by 2 years), were 1 year longer since the primary, were followed up for 3 years less, and were more likely to have uncemented or hybrid implants. In discussion they state: "Findings in this study should be interpreted in the light of residual confounding not known to us" But there is clear confounding that is known to them, as well as other confounding that may not be - they should discuss this in the discussion section

From Figure 1 it does not look like the number of one-stage procedure has been increasing in the 'last' years - this statement is erroneous

Author Response

Point 1: In line 82 what do they mean when they say: "Therefore, this data was not made available for all revision cases identified for the current analysis". The fear in this study is that a significant number of cases were missed out of the analysis (as many as 25% in certain periods of the study. This could make a big difference to the results.

Response 1: ASA score and BMI were first registered in the Swedish Hip Arthroplasty Register in 2008. As commented, this means that this data could not be retrieved for all cases during the entire study period. However, we chose to perform a subanalysis on cases from 2008-2015, as BMI and ASA were available for most of these cases, to investigate whether there was a difference in outcome when these variables were adjusted for. Despite these adjustments, our results remained the same for this group as well, as no statistically significant difference in outcome was found. We have clarified this in the revised manuscript (p. 2, line 82-83, 90 and p. 6, line 150-152).

Point 2: In their Stratification, its clear that the two groups are not the same - two-stage revisions were younger (by 2 years), were 1 year longer since the primary, were followed up for 3 years less, and were more likely to have uncemented or hybrid implants. In discussion they state: "Findings in this study should be interpreted in the light of residual confounding not known to us" But there is clear confounding that is known to them, as well as other confounding that may not be - they should discuss this in the discussion section

Response 2: As the stratification showed, patients selected for the one-stage procedure were older. This may support one theory that we have discussed regarding the selection of patients for either procedure. Some surgeons may feel that the one-stage procedure is better for elderly patients to avoid making them subject to further procedures. However, the evidence for age as such as a risk factor for infection is inconsistent and has therefore not been adjusted for. We cannot explain the difference in mean time from the primary procedure to revision for infection in patients treated with the two-stage procedure. More part of the one-stage procedures were performed early on in the study period, allowing for a longer follow up time in this group. Also, the one-stage group were revised one year earlier than the two-stage group, which may have added on to the longer follow up time. The combination of the two aforementioned factors may explain why the two-stage group were followed for a shorter period of time. Patients in the two-stage group were more likely to have uncemented implants as younger patients usually receive uncemented implants, and older patients usually receive cemented implants. Previous research has not been able to show what role type of cementation has in regard of PJI. We have added this to the discussion in the revised manuscript (p. 7, line 191-201).

Point 3: From Figure 1 it does not look like the number of one-stage procedure has been increasing in the 'last' years - this statement is erroneous.

Response 3: This statement was based on the percental increase between the years 2010-2015 compared to 2000-2009. We realize that this was not explained properly, and that this was difficult to interpret from the table and figure based on our current statement. Therefore, we decided to revise the statement to clarify that the one-stage procedure has proportionally increased during 2010-2015 compared to 2000-2009 (p. 3 line 110).

Round  2

Reviewer 1 Report

Studies by Gromov et al in Clin Orthop Relat Res 2015 corresponding to Danish Registry and Karachalios et al in EFOR Open Res 2018 may be relevant.